# Digital data storage on DNA tape using CRISPR base editors

Afsaneh Sadremomtaz[1,3], Robert F. Glass[2,3], Jorge Eduardo Guerrero[1,3], Dennis R. LaJeunesse[2], Eric A. Josephs [2] ✉ & Reza Zadegan [1] ✉

While the archival digital memory industry approaches its physical limits, the demand is significantly increasing, therefore alternatives emerge. Recent efforts have demonstrated DNA's enormous potential as a digital storage medium with superior information durability, capacity, and energy consumption. However, the majority of the proposed systems require on-demand de-novo DNA synthesis techniques that produce a large amount of toxic waste and therefore are not industrially scalable and environmentally friendly. Inspired by the architecture of semiconductor memory devices and recent developments in gene editing, we created a molecular digital data storage system called "DNA Mutational Overwriting Storage" (DMOS) that stores information by leveraging combinatorial, addressable, orthogonal, and independent in vitro CRISPR base-editing reactions to write data on a blank pool of greenly synthesized DNA tapes. As a proof of concept, this work illustrates writing and accurately reading of both a bitmap representation of our school's logo and the title of this study on the DNA tapes.

As digital information production grows exponentially and the industry approaches physical limits, high-density long-term storage solutions are necessary[1,2]. DNA as an alternative to the archival storage medium offers several potential advantages, including higher density and retention, and lower energy consumption compared to the state-of-the-art memory materials[3–6]. De-novo DNA synthesis has enabled the development of DNA-based memory technologies[7–10] and the 2018 Semiconductor Synthetic Biology (SemiSynBio) Roadmap predicts that the speed and cost of DNA synthesis will improve dramatically in the future. Additionally, automation of writing and reading processes may improve the portability and scalability of DNA memory. However, the widely used de-novo chemical DNA synthesis methods are unlikely to meet the scalability required for high-volume memory manufacturing, and generate massive amounts of toxic waste, which is not sustainable[11–16]. For context, our calculations demonstrated that storing 5 min of a 1080p YouTube video stream in commercially acquired DNA costs over $ 7 million dollars, consumes over 100KWh of energy, takes over 4 days, and produces over 15 liters of toxic waste

(Supplementary S1). This implies, to store every bit of information generated by 2030 (~6e + 23 Bytes) in synthetic DNA, the current technology would produce nearly 85 petaliters of hazardous waste, which surpasses the volume of water that the Mississippi River empties into the Gulf of Mexico over 40 years[17]. While the majority of academic and industrial efforts target scalability, the DNA memory community has mostly ignored the disruptive environmental consequences of large-scale de-novo DNA synthesis. Although a few recent efforts[18–21] have been focused on the enzymatic or green synthesis of DNA, writing information in packets of newly synthesized DNA each time a new data is being stored is unlikely to satisfy the expected scalability for a storage medium. For DNA-based memory innovations to become mainstream environmental and scalability concerns need to be addressed.

Moreover, one of the advantages of semiconductor data storage systems is the scalability due to the large manufacturing of blank medium and on-demand information storage on the blank units. Our molecular data storage system, which we call 'DNA Mutational Overwriting Storage' (DMOS), uses combinatorial, addressable, orthogonal,

[1]Department of Nanoengineering, Joint School of Nanoscience and Nanoengineering, NC A&T State University, Greensboro, NC, USA. [2]Department of Nanoscience, Joint School of Nanoscience and Nanoengineering, UNC Greensboro, Greensboro, NC, USA. [3]These authors contributed equally: Afsaneh Sadremomtaz, Robert F. Glass, Jorge Eduardo Guerrero. ✉e-mail: eajoseph@uncg.edu; rzadegan@ncat.edu

and independent in vitro CRISPR base-editing reactions to overwrite data on the pre-defined domains of existing DNA. Akin to conventional magnetic tape architecture, the 'blank DNA tape' consists of multiple DMOS registers and each register contains a set of 16 domains (bits) –including 'state' and 'index' sections. We employ CRISPR base editing[22] to write the information by mutating the sequence of a 'state' section from the unmutated state (0) to the mutated state (1). Using nanopore sequencing, we recover the 'mutational signature' of each DNA tape register that informs our error-correction and coding schemes to retain the data accurately and precisely. As a proof of concept, we wrote 1250 bits of data, including a bitmap representation of our school's logo and the title of this paper on multiple DNA registers, and recovered the stored data with 100% accuracy. This work is the first demonstration of writing digital data in the form of sequence edits at precise locations of a pre-existing blank pool of DNA tapes produced that were greenly synthesized *en masse* via replication in bacteria.

## Results

### The writing mechanism

To write the data on DMOS bits, we developed a programmable molecular writer system that uses a CRISPR base-editing reaction. During the base-editing reaction, the CRISPR effector Cas9 first recognizes and binds to a 3 bp sequence known as the protospacer adjacent motif (PAM) and a 20 bp-long sequence complementary to its RNA cofactor (its targeting guide RNA or gRNA). The 20 bp gRNA-complementary sequences within the double-stranded DNA registers make up the 'state' sections of each domain (Supplementary Figs. 1A, S1, S2). After binding by the CRISPR effector, the resulting complex forms a nucleotide structure known as an R-loop. During CRISPR-Cas9 gene editing, the formation of the R-loop would trigger the Cas9 to generate a double-strand break in the DNA molecule. However, in a modified base-editing reaction we use a mutant form of Cas9 known as dead Cas9 (dCas9) that still recognizes and binds to specific DNA targets to form an R-loop but keeps the DNA intact[23] (Supplementary S2, Fig. S1). We then introduced a mutagenic protein APOBEC3A to modify the state sequences at highly-susceptible displaced DNA strands within the R-loop independently at their targeted positions. These reactions mutate deoxycytidine (dC) to deoxyuracil (dU), which is subsequently converted permanently to deoxythymine (dT) in the target domains with a resolving biochemical reaction (Supplementary S2, Fig. 1A). APOBEC3A efficiently mutates dC's in single-stranded DNA or R-loops, therefore only the dCs in registers with a bound dCas9 will be mutated, while those in registers without dCas9s will remain unmutated. Indeed, analysis of targeted versus non-targeted bit sequences confirmed that APOBEC3A only affected DNA when bound by dCas9. The other state segments remain base-paired, and therefore the mutations happen only in the displaced strand of an R-loop of a targeted bit (Fig. S1).

We designed and experimentally validated a set of 16 unique state sequences where (i) Cas9 exhibited robust double-strand cleavage activity at the target location (Fig. S2), indicating that each Cas9-gRNA ribonucleoprotein (RNP) could form a stable R-loop with its target state sequence[24]; (ii) each displaced DNA sequence contained at least two dTdCdR nucleotide motifs[25] that are high-activity substrates of APOBEC3A (where dR is a dG or dA nucleotide); and (iii) those motifs were located in a mutagenic hot-spot that we identified positioned at least 6 nt away from the PAM region (Fig. S3). The last criterion is a result of our finding that dC's located close to the PAM (within ~6 bp) exhibited significantly lower mutation rates (Fig. S3). To write the data, the dC's across the state section of a DMOS bit are mutated to dT's if and only if Cas9 RNP targeting the state segment of that specific bit is included in the reaction (Fig. 1). To read the state of the bit we determined whether dCs in that bit were mutated (1) or unmutated (0).

### The architecture of DMOS blank tape registers

Like writing data onto blank tapes, the DMOS overwrites the sequence (state) of the DNA domains to write the data. Each DMOS tape contains several registers, and each register consists of the 16 different domain bits (Figs. 1A and 2A). Each domain bit consists of a 23 bp-long 'state' section (PAM and sequence recognized by one of the gRNAs) and a unique 40 bp 'index' sequence (Figs. 1A, 2A). This architecture spaces out the bits far enough for compatibility with the writer system, resulting in independent interactions and reduced crosstalk. We used the indexes to localize a bit along a register during sequencing, to increase error-tolerance, and later to help in diversifying the pool of registers to generate a DMOS block with registers that could be easily differentiated in one single sequencing run to reduce sequencing time and costs.

We tested three addressing methods to differentiate and order the different registers (Supplementary S4) to increase data capacity in a single sequencing run. For the first method, we used a PCR-based addressing scheme that exploited the 36 combinations of 6 unique barcoded primers to differentiate the registers in the pool (Fig. 2B). In our second method we developed a coding scheme to permute the positions of each of the 16 common domains in a register according to its address in the data block (Figs. 2B and S4), the unique order of which could then be used to determine the identity of that individual register in a process we termed "domain-calling". To construct the permutations used for addressing the registers, we used two schemes: a lexicographic addressing scheme and high-entropy addressing scheme. In the lexicographic addressing scheme, the positions of the domains in the first N registers of a block were permuted 'in alphabetic order' (Fig. 2B); however in our specific work this turned out to be more error-prone since sequential registers tended to be very similar, and this similarity could lead to incorrect domain-calling and addressing. The high-entropy addressing scheme was designed so that the degree of difference in the permutation of domain orders between sequential registers were significantly higher using a deterministic shuffling scheme[26]. Therefore, once the order of domains of a register was determined from sequencing, we could still map that unique ordering to its intended register address efficiently (Fig. 2B). Using these addressing schemes, we created a DMOS block (tape) of 48 registers, each containing a unique permutation of the 16 common domains (Fig. 2B) that, after initially created via chemical synthesis, were cloned into plasmids and continually replicated in bacteria as needed.

### The orthogonality and modularity of the DMOS system allow for efficient bit-level writing and reading

We tested whether the modular DMOS writer system using dCas9 RNPs/APOBEC could independently transduce the digital data to the corresponding mutational domains across the registers of the DNA tape at the same time (Fig. S4). We screened the cross-reactivity of the writer system by determining the by determining the mutation rates (dC to dT) of individual domains for each of the 16 common domains on a single register. As shown in Fig. S4A, we confirmed that multiplexed targeting of neighboring domains did not change the mutation rate or mutational signatures compared to those domains mutated one-at-a-time (Figs. S3B, S4, and S5). Therefore, the domain-level mutation was found to be independent and orthogonal, allowing the data for each register to be written all at once by adding the required CRISPR RNPs.

The observed domain-level mutation rates and mutational signatures also informed our algorithm to convert nanopore sequencing reads to digital data from the presence or absence of the observed mutational signature using a Bayesian classifier (Figs. S1, S4, and S5). Observed domain mutation rates that were significantly higher or lower in comparisons to the thresholds determined by earlier validating experiments (Table S2) were assigned 1 or 0, respectively, with

high confidence while those near the threshold were called uncertain (Figs. S4 and S5) and subjected to additional and more stringent Bayesian analysis of their read sequences to infer their state based on that bit's mutational signature (the precise sequence and position of dC's mutated). Overall, our results highlighted orthogonality, independence, and reproducibility across bits of DMOS registers (Supplementary S5). We believe, sources of error could include variations in the activity of the writing enzymes, particularly if the activity causes the expected mutation rate or signature to differ significantly from the training sets, or if registers were mis-addressed, which happened significantly less frequency when high-entropy permutation was performed.

To test our ability to perform writing, addressing, and reading of a series of specific bits in order on a DMOS tape, we created a bitmap representation of our school's name (Joint School of Nanoscience and Nanoengineering) in 512 pixels across 32 DMOS registers (Fig. 3). Delivery of specific CRISPR RNPs to the different registers in a 96-well plate were automated using an open-source Opentrons robot and determined by the locations of the 0 s and 1 s. in the bitmap. After processing and sequencing the DMOS tape, we performed the two-stage Bayesian classifiers to differentiate mutated and unmutated states from every sequencing read assigned to each DMOS register (Fig. S6). After only 20,000 sequencing reads, we could partially recover the intended bitmap with 97.7% (500 out of 512

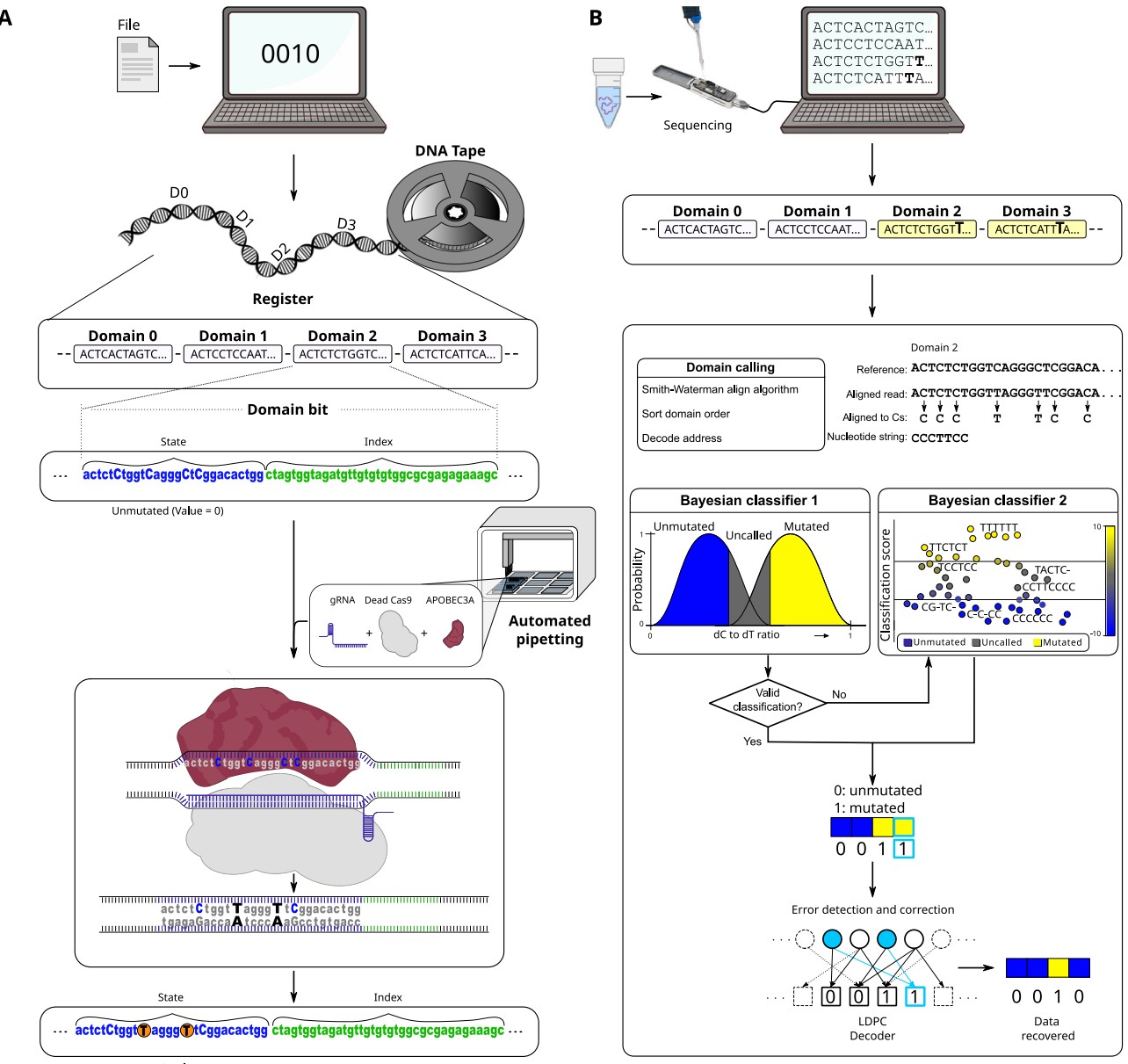

**Fig. 1 | Schematic view of writing data onto DMOS DNA tape.** To write and read the data the encoder uses binary files and converts them into byte arrays with codewords. The coding protocol informs the writer system and determines the desired edits on the state (bit) sections of the DNA registers resulting in changing the states of bits from 0 to 1 on DNA tape. **A** Writing data on blank magnetic tape. Input data are converted to the binary message and inform the mutation process. DMOS uses a programmable molecular writer system that drives CRISPR base-editing reactions. CRISPR/dead Cas9 (dCas9) accompanied by APOBEC3A drives mutation reactions for the parallel rewriting of data in state sections of DNA tape registers. **B** To read the data, we use nanopore DNA sequencing. The output of sequencing reads is decoded by first performing a local alignment (Smith–Waterman alignment) followed by a Bayesian analysis to determine the mutated states and convert the data back to the binary form.

**A**

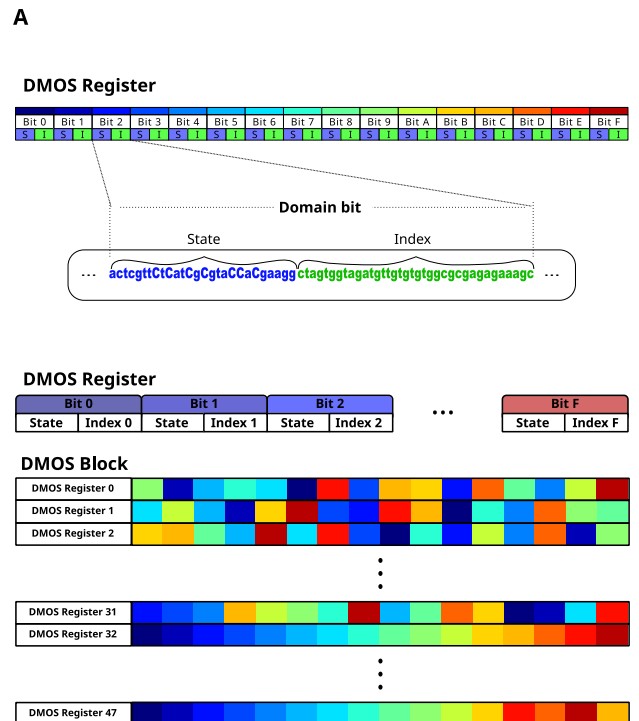

**B**

Fig. 2 | DMOS blank DNA tape. DMOS register defines as a set of 16 pre-determined domain bits. **A** A single bit defines as the 'state' section with 23 bp length and a 40 bp 'index' sequence. To increase data storage capacity, we defined a unique shuffling (permutation) addressing model for the domain position orders (bits) in each DMOS register that generates 48 different combinations of DMOS register as we called "DMOS data blocks" or a "DMOS tape". **B** Designing address schemes of the DMOS register. Demonstration of DMOS registers from a trace pool once passing through the nanopore that differentiates three different addressing schemes, including the Barcoding scheme, Lexicographic permutation, and High-entropy permutation addressing scheme.

correct) accuracy: with 5 false 'unmutated' (0) bits, 1 false 'mutated' (1) bit, and 6 bits that could not be called with high confidence. We assessed the robustness of the method via bootstrap analysis of 250 randomly generated sequencing data streams, where the data streams containing a fixed number of reads were randomly selected with replacement from the full sequencing dataset. Notably, the 'bit recovery' curve reaches a plateau after the first 10,000 reads with a recovery rate of 91.37 ± 1.7% standard deviation (Fig. 3), with only a small increase in additional data recovery in the subsequent 10,000 reads (Figs. 3 and S6). This finding justified the implementation of error-correction codes to achieve acceptable data recovery rate for memory purposes with limited computational and sequencing costs.

## The trade-off between writing overhead and reading recovery rate

Informed by previous reports[27] and our statistical bootstrapping analysis, and to reduce the overhead costs of the system, we studied the trade-offs of the writing overhead, or amount of added data redundancy that ensures data recovery during the reading process, and recovery rate. We employed a strategy that accounts for addressing errors and domain-specific error rates and error-correction codes while maximizing the bit capacity of the registers to match the domain length of the proposed architecture: low-density parity-check (LDPC) codes that set the error threshold close to theoretical Shannon capacity limits. The code includes a pre-processing step that adds LDPC error correction to the digital binary contents. Leveraging our bitmap representation sequencing data, we ran an error-recovery simulation for different LDPC codes and generated the models of the error-recovery rate for each decoder[27]. We generated 1000 random data streams and evaluated error-recovery rates of each decoder (Fig. S7). Then, we created a library of Protograph and Regular LDPC codes that

we used in the subsequent studies (Figs. S7, S8A, and S8B). We found that with greater amounts of writing overhead, the data recovery will be faster and more accurate with fewer sequencing reads (Supplementary Figs. S7, S8A, S8B, and S9). However, we note that the block's storage capacity—and, hence, the cost of wet-lab components—are also affected by increasing redundancy (Fig. S4A). Therefore, we concluded that adding 25 percent redundancy results in sufficiently accurate data reconstruction while keeping the overall overhead at a minimum (Figs. S7 and S8A).

## Writing and reading digital information on DNA tape

We wrote the title of this article ("Digital data storage on DNA tape using CRISPR base editors") in ASCII in one DMOS tape block containing 48 registers (Fig. 4). We developed a semi-automated coding platform that utilizes the Protograph LDPC code with 25 percent redundancy. The coding algorithm generated a file with the list and addresses of required mutations and communicated with the Opentrons robot to conduct the targeted base-editing reactions on the 48 DMOS registers (Figs. 4 and S5). We then performed nanopore sequencing, Bayesian classification, and decoding to read the DMOS tape and retrieve the message.

We evaluated and normalized the probability of the mutations for each isolated DMOS bit and identified their mutational signature profiles (Fig. S10). We recorded the results in real time and converted the sequences to binary data, translated it to a text file during 100 read intervals, and evaluated the recovery of the file. Based on our calculations, we expected that our decoder will be able to recover the file when the data stream includes 20,000–100,000 reads. After 100,000 reads, the read message was 96.7% correct with only 25 errors in 768 bits, and using the decoder algorithm we fully restored the intended message (the title of this article) with 100% accuracy and completeness (Fig. 4).

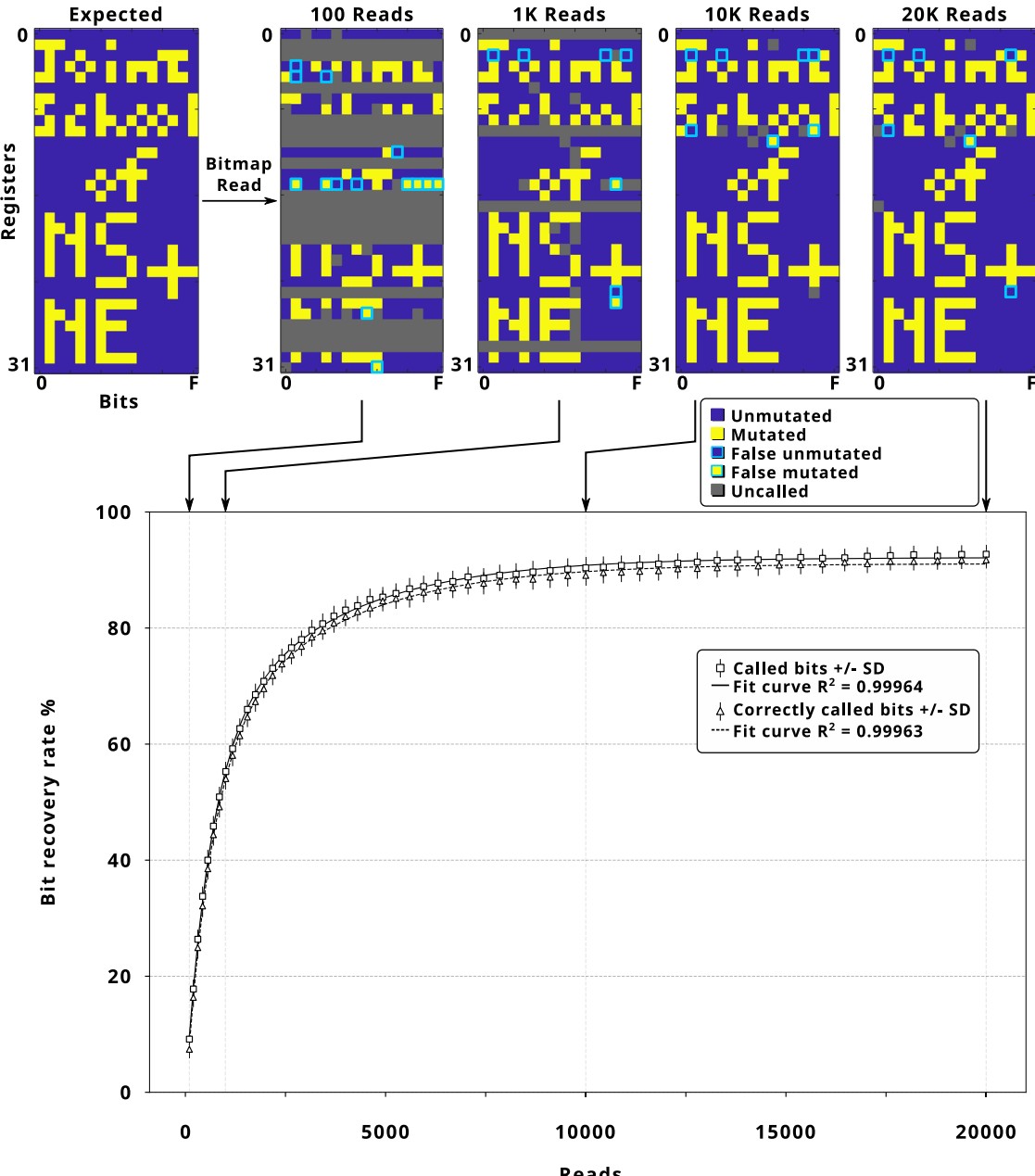

**Fig. 3 | Bitmap representation of the logo of our school (the Joint School of Nanoscience (NS) and Nanoengineering (NE)) written on DMOS tape.** The DMOS decoder records snapshots in every 100 nanopore sequencing read. Here, an example of these snapshots at 100, 1000, 10,000, and 20,000 reads is presented. The bootstrap analysis was performed with 250 replicate selections from different positions, which is represented as the recovery rate of called bits and correctly called bits vs. reads. Data are represented as average values ± SD. Source data are provided as a Source Data file.

As before, we found the rate of improvement for the data recovery process slowed after ~22,000 reads (87.37 ± 0.62% standard deviation). To optimize the performance of DMOS reading, we ran multiple simulations to assess the performance of several Protograph and Regular LDPC codes with different redundancy percentages to inform future studies (Supplementary S8, Figs. S7 and S8). For example, by using Protograph LDPC error-correction with redundancies of 33 and 50 percent, the decoder would have been able to fully recover the intended message after 50,000 and 3000 reads, respectively (Figs. S7 and S8C). Overall, we showed that through implementing optimized encoding strategies, we are able to permanently write hundreds of bits of digital data into DNA molecules via targeted mutation with modular CRISPR RNPs on-demand, but without the need for DNA synthesis.

## Discussion

'DNA Mutational Overwriting Storage' (DMOS) tape leverages the architecture of semiconductor memory devices and recent developments in gene editing technologies to write digital data in the form of precise DNA sequence edits on pre-made DNA molecules. After initial synthesis, these universal DNA tapes registers can be copied indefinitely, e.g. using bacteria[3,7–10,13], therefore DMOS has the potential to bypass some challenges of conventional DNA-based data storage, including the generation of toxic waste and scalability (Fig. S1)[28–35]. When DNA is chemically synthesized (in most DNA writing systems), the operation needs access to DNA synthesizer machines in order to write a new data and editing of the recorded data requires the synthesis of the whole DNA pools or significant portions of it. Here we performed DMOS-based encoding using commercially-available

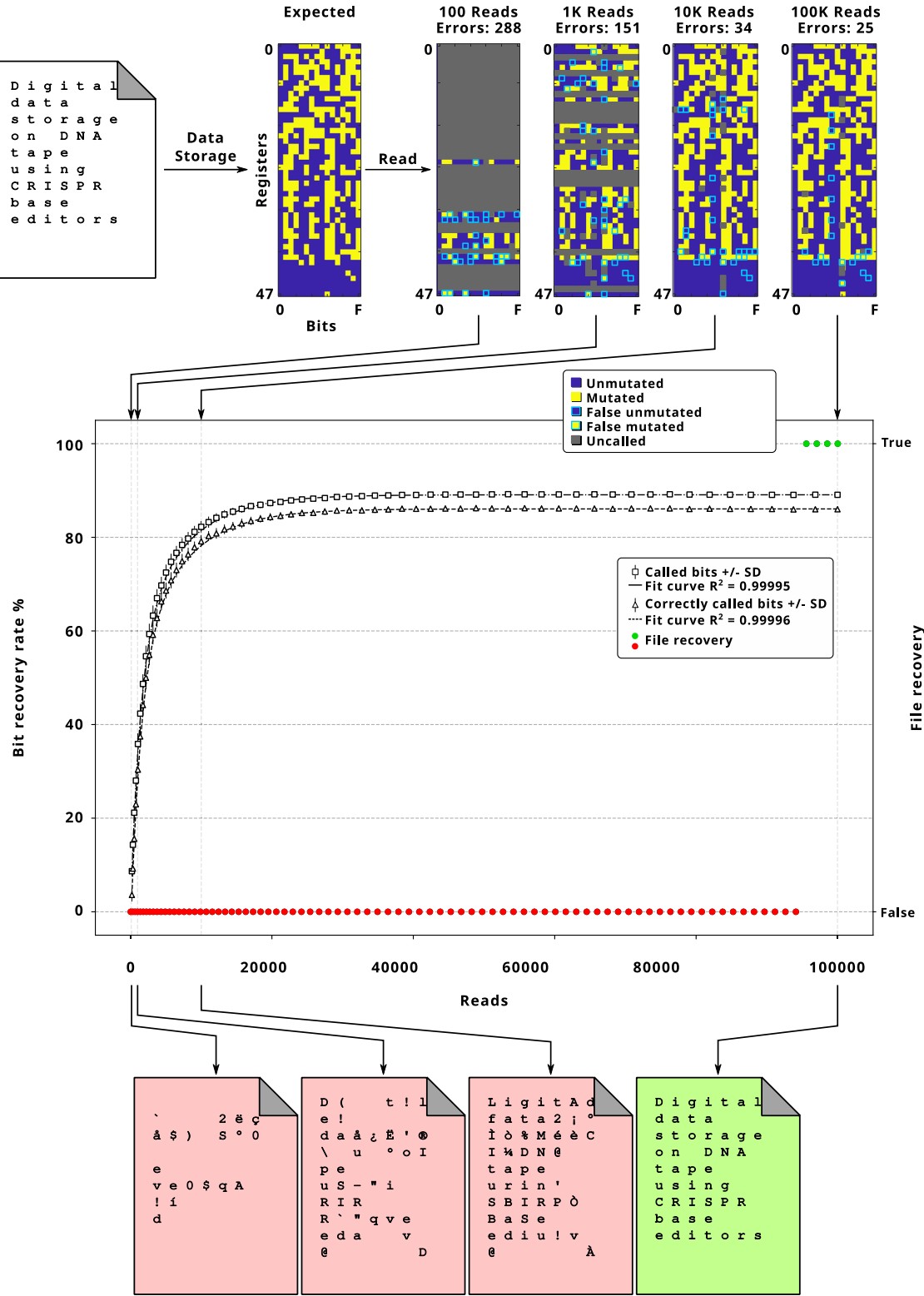

**Fig. 4 | Writing the title of this study on DMOS tape.** We converted the title of the paper from ASCII to binary, added error-correction, performed the mutation experiments, sequenced the DNA blocks, and recovered the data. DMOS decoder captured the snapshots of sequencing data streams every 100 intervals. A boot-strap analysis was performed using 250 randomly generated data streams derived from DNA sequencing data, with replacement selected randomly from the full dataset. The graph represents the DMOS recovery rate for called bits, correctly called bits and file recovery vs. reads. Data are represented as average values ± SD Source data are provided as a Source Data file.

enzymes, and as we discuss below, the use of common domains in the DMOS registers provides opportunities for rewriting or editing data within the tapes.

Since the majority of these challenges are due to the need for de-novo DNA synthesis, a few efforts have focused on alternative DNA-based storage media[11,30,36–38]. To date, a few reports have provided more sustainable solutions for DNA synthesis, including enzymatic DNA synthesis[10,12,18–21]. Some recent studies focused on information stored in DNA nanostructures with rewriting capabilities[39–43], for example 56 bits of data have been stored using hairpins and overhangs on a long single-stranded DNA genome of M13 while reported retention of 90 percent[30]. In another report, 20 bytes (160 bits) of digital data was stored in DNA nanostructures and read back using super-resolution microscopy with 100 percent data retention[39]. Another study stored 14 KB of data in the form of DNA backbone nicks of pre-existing DNA hard drives[13,31]. While promising, the majority of these efforts lack scalability or sustainability, and because they are often based on epigenetic or structure-based changes to the DNA molecule, they are not easily copied using principles of DNA replication as DMOS tapes are.

While information density at the per-nucleotide level does not compare to synthesis-based DNA data storage methods, DMOS provides new opportunities for scaling the system, expanding data storage capacity, and data rewriting. For example, synthesis-based methods typically require high-accuracy sequencing methods that have limited read lengths (typically at a few hundred bp), while DMOS can make use of long-read nanopore sequencing, which is less accurate but can read lengths of 10's to 100's of kbp. As more DMOS protospacers containing sequences that are capable of both robust CRISPR recognition and base-editing activity are identified or engineered, DMOS DNA data tapes can be made significantly longer[11–16]. Furthermore, the spectrum of mutations can be expanded to increase the number of possible "states" a domain can be classified as, e.g. there are numerous APOBEC and adenosine base editor (ABC) orthologs[44], each with different mutational signatures and propensities to mutate dC's to dT's or dA's to dG's in different sequence contexts. These different mutational signatures could be differentiated through the DMOS reading process to further increase information density, and if the base editor enzymes are tethered directly to the dCas9 enzyme prior to RNP formation, different base editors could be used simultaneously to give each domain a different mutational signature. In that case, because CRISPR effectors tend to bind effectively irreversibly to their DNA targets in vitro[45], each domain could also be targeted simultaneously with different combinations of base editors, and the presence of multiple mutational signatures. That would allow up to $2^n$ possible states could be identified per domain, where $n$ is the number of potential base editors. Additionally, if PAM sequences were positioned on both the top and bottom strands of a domain and both the pro-tospacer and its reverse complement could be robustly recognized by a dCas9, then even the same base editor would have a different mutational signature, whether targeting the top or bottom strand.

Overall, in principle the information storage capacity of a single standardized DMOS register with multiple domains could potentially scale, using combinations of modular components derived from a finite set of gRNAs and base-editing RNPs. We also note that the ability to target both the top and bottom strand of a domain and using both ABOPEC and ABCs allows for the possibility of erasing and rewriting data from DNA tapes. For example, after some dC's converted to dT's (on the top strand) on one strand during an initial data recording reaction with APOBEC to flip a "0" to a "1", dAs paired with the new dTs could be subsequently converted back to dG, which would revert the dT back to dC and flip the bit back from "1" to "0" (Supplementary Fig. S1B). Rewriting data that is already written using synthesis-based approaches is potentially challenging. With regards to cost of the DMOS system, an advantage is that once the tapes have been initially synthesized, they can be reproduced infinitely. The biomolecules used for enzymatic writing can be produced at scale, and the long-read sequencing used in DMOS for reading is significantly less expensive than other types of next-generation sequencing that other forms of DNA data storage typically require. Lastly, by incorporating CRISPR prime editing[46], where portions of a gRNA are reverse transcribed and directly integrated into a targeted DNA sequence, in a reconstituted reaction could further increase the capabilities of writing or rewriting data onto DNA tapes in a "synthesis-free" manner using the DMOS system.

## Methods

### DMOS register synthesis and cloning
The DMOS registers are made up of DNA sequences that have at least two dTdCdR sites (R = Purine, A or G). These sites have been proven to be effective binding sites based on the research conducted by Chari[24] and synthesized by TWIST Bioscience. To facilitate the mass production of DMOS registers, we assembled the registers into the pBR322 plasmid (New England Biolabs) using the NEBuilder© HiFi Assembly Master Mix. We inserted the register into the plasmid by cleaving the pBR322 plasmid using FastDigest restriction enzymes Bsu15I and EcoRI.

Next, we transformed the modified plasmid into NEB 5-α competent *Escherichia coli* bacteria and grew them under Carbenicillin antibiotic resistance. To extract the assembled plasmid, we used the Monarch Plasmid Miniprep Kit (NEB).

### sgRNA synthesis
We chose the DNA sequences of DMOS bits that demonstrate effective binding sites, relying on the research conducted by Chari[24]. The coding oligos were procured from Integrated DNA Technologies (Table S3). These oligos were used for generating the 16 gRNAs that target the modular domain state sequences. Each sgRNA was synthesized using the *Streptococcus pyogenes* EnGen© sgRNA Synthesis kit (NEB) with 1 µM concentrations of oligo bits. The synthesized RNA was purified using RNAClean XP magnetic particles (Beckman Coulter), and concentrations were measured using a Nanodrop Lite Spectrophotometer (Thermo Scientific).

### Enzymatic writer protocol
dCas9 RNPs were formed by mixing 1 µL of 10× Cas9 buffer [200 nM HEPES, 1 M NaCl, 50 mM MgCl2, 1 mM EDTA, pH 7.4], 1 µL dCas9 (1 µM), 1 µL DNA template (50 nM) in a 10 µL reaction volume and incubating at 37 °C for 1 h. Targeted deamination of the DMOS tapes was performed by first dispensing 1.5 µL of the desired dCas9 RNPs and 1.5 uL of the DNA (50 nM) then adding the deamination master mix (8.5 µL of nuclease-free $H_2O$, 1.5 µL of 10× Cas9 buffer, 1 µL (40 units) of RNAse Inhibitor Murine, 1 µL APOBEC (8.7 µM) and 0.5 µL of BSA (from the NEBNext© Enzymatic Methyl-seq Conversion Module kit from New England Biolabs)) to a final volume of 15.5 µL, centrifuged briefly, and incubated at 37 °C for 3 h.

To stop the reaction, 1 µL (0.8 units) of Proteinase K was added and the reaction was incubated at 56 °C for 10 min, then purified using AMPure XP magnetic beads following standard protocols to 20 µL. After eluting the sample from the beads, the DNA was treated with Lambda exonuclease to degrade the unmutated strand by adding 5 µL of 10× Lambda Exonuclease buffer and 1 µL of Lambda Exonuclease in a 50 uL reaction volume and incubated at 37 °C for 30 min, followed by heat-inactivated at 75 °C for 15 min then purified using AMPure XP magnetic beads. The samples were amplified in a PCR reaction using Q5U polymerase (NEB) with an annealing temperature of 63 °C and an extension of 72 °C at 45 s. The schematic view of this protocol is depicted in Fig. S1.

### Sequencing run and basecalling
Sequencing was performed using an Oxford Nanopore MinION Mk1B nanopore sequencer supported with the MinKNOW software. We prepared a library of DMOS registers with stored files using the ligation

sequencing kit "LSK-110" and the sequencing run was carried out on R9.4.1MinION Flongle flow cells from Oxford Nanopore Technologies at default settings on MinKNOW. The fast5 raw signal files were base-called using Guppy basecaller 6.1 for high-accuracy basecalling on a laptop with Alienware m15 R4 1TB SSD with an Intel i7 10750H CPU, 16 GB of RAM and dedicated NVIDIA GeForce RTX 3060 GPU in the super high-accuracy (sup) mode. The generated FASTQ files were binned into pass or fail folders based on their q-scores. Only the reads that have passed the q-score threshold were analyzed.

### Orthogonality tests on DMOS template

We tested the orthogonality of the DMOS writer system following these conditions: (1) Including dCas9, APOBEC3A, and gRNAs that targeted all 16 DMOS bits simultaneously across the register (triplicate); (ii) no dCas9, no APOBEC3A (triplicate); (iii) including dCas9, APOBEC3A, and gRNAs targeting individual DMOS bits per the DMOS register (triplicate). The mutation reaction performed for all experimental and control reactions under the same standard mutation condition. Next, the reaction was stopped individually with 1 uL of Proteinase K (0.8 units) and purified using AMPure XP magnetic beads accordingly. The purified samples were treated with Lambda exonuclease and purified using the standard protocol. Each reaction was amplified and purified individually, and an equimolar from each sample was taken and combined together for the nanopore sequencing. Barcoded primers were used for addressing of DMOS register in this experiment and listed in Table S4.

### Software development

We developed our DMOS D-coder using Python language and the Spyder IDE. The error-correcting layer uses the Protograph LDPC library (https://github.com/shubhamchandak94/ProtographLDPC)[47–50]. To design our LDPC code, we selected the Protograph type accumulative repeat by 4 jagged accumulate to define the Generator and Parity-Check matrices, with a message-code ratio of 3/4, expansion factor 96. These parameters constructed an LDPC code that uses 576 bits per message (72 byte) and 768 bits per codeword (96 bytes). We developed a Python script to communicate with the LDPC library that allows the conversion of the intermediate binary files for input/output and capture the diagnostic signals of the LDPC decoder.

The DMOS software layer uses two main modules to retrieve the binary file: DMOS decoder and LDPC decoder. The DMOS decoder was written in C++ using the QtCreator IDE, and uses the Smith–Waterman algorithm (https://github.com/mbreese/swalign) to align DNA sequences[51]. We list all the threshold values used in the first Bayesian step (Table S1), and the second Bayesian step uses trained data available in the code repository. We created a graphical user interface using PyQt5 to easily select the input samples and configure the parameters for the DMOS decoder. We developed the simulation scripts in Python language and used standard libraries such as Numpy, Matplotlib, and statistics.

### Automation of writing via OT-2 pipetting robot

We used the Opentrons OT-2 pipetting robots for the automated data writing procedure into DMOS tape. This procedure requires the following plate preparations: We reserved one plate for the dCas9 library of the 16 mutational bits, the second plate contains individual blank DNA registers with addresses; located in separate wells, and the last plate includes a rest master mix content. We developed Python scripts for customizing the mutational list file map and delivering it to the following steps: First, the robot locates the target registers (Table S4) in separate pools. Next, it takes volumes of 1.5 μL from the dCas9 library plate and mixes them in the master mix plate. This step is followed by taking 1 μL from the master mix and depositing it into the selected register pool. Finally, we incubated the reaction at 37 °C for 1 h at the thermocycler. The samples were addressed for the clean-up step

in which we used multichannel tip robots to deposit 30 μL of AMPure XP beads into our mixture before activating a magnetic rack for 2 min. Next, the magnetic rack was engaged, the supernatant discarded, and the beads were eluted with an elution buffer. The pure DMOS registers pool moves forward with nanopore sequencing.

### Encoding of a file onto DMOS register using DMOS writer system

Following the predetermined map of the mutation list to write the data on registers (Fig. 4), the dCas9 RNP pool was prepared and using the Opentrons pipetting robot distributed on the corresponding DMOS registers. Each RNP mixture had a final concentration of 50 nM to preserve the 10 to 1 RNP to register ratio in every single reaction. Next, the APOBEC3A was added to each reaction and incubated for 3 h at 37 °C inside a veriflex Thermocycler. The reactions were stopped using 1 uL of Proteinase K, to degrade the dCas9, and purified using AMPure XP beads. The purified samples were each treated with Lambda Exonuclease and purified. The purified samples were amplified following this PCR setting; primary denaturation 98 °C for 30 s, denaturation 98 °C for 10 s, annealing 63 °C for 20 s and an extension 72 °C for 45 s for 30 cycles followed by final extension at 72 °C for 2 min. The registers were purified using standard AMPure XP bead protocol. The samples were combined and sequenced using a nanopore sequencing.

### Reporting summary

Further information on research design is available in the Nature Portfolio Reporting Summary linked to this article.

## Data availability

The analysis data generated in this study have been deposited in the DMOS Figshare database under the link https://doi.org/10.6084/m9.figshare.24143649. The raw data generated in this study have been deposited in the GitHub repository under the link https://github.com/SBMI-LAB/DMOS_data with DOI 10.5281/zenodo.8347270. The Sequenced data generated in this study has been deposited in the NCBI database under the accession code PRJNA1022044. Source data are provided with this paper.

## Code availability

The code for DMOS Encoder generated in this study have been published in the GitHub repository https://github.com/SBMI-LAB/DMOSEncoder with DOI 10.5281/zenodo.8347315. The code for DMOS Decoder generated in this study have been published in the GitHub repository https://github.com/SBMI-LAB/DMOSDecoder with DOI 10.5281/zenodo.8347307.

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

## Acknowledgements

We thank our colleagues Shyam Aravamudhan, Daniel Herr, and Micheal Brandon Reed for their support and suggestions. This work is funded

through NSF SemiSynBio-II: DNA Mutational Overwriting Storage (DMOS) (award # MCB 2027738) and in part supported through NIH award # R35GM133483. This work was performed at the Joint School of Nanoscience and Nanoengineering, a member of the Southeastern Nanotechnology Infrastructure Corridor (SENIC) and National Nanotechnology Coordinated Infrastructure (NNCI), which is supported by the National Science Foundation (award # ECCS-1542174).

## Author contributions

A.S., R.G., and J.G. have equally participated in performing experiments, analyzing the data, and writing the manuscript. D.L. participated in writing and initial idea development. E.J. and R.Z. supervised the work and participated in writing.

## Competing interests

The authors declare no competing interests.
