## [Peer Review File · Nature Communications]

Reviewers' Comments:

Reviewer #1:

Remarks to the Author:

In this manuscript, Sadremomtaz et al. propose a DNA-based data storage architecture named 'DMOS', which uses dsDNA fragments with barcoded "domains" and a guided base editor (which is a dCas9+APOBEC3A) to record data. Basically, the enzyme fusion is guided towards a predetermined target sequence, and switches dC bases to dU and subsequently dTs after a resolving reaction (an exonuclease treatment followed by PCR). The recorded information is then read out by nanopore sequencing. The method is then used to record 1250 bit of data. While the design and experimental procedure are technically sound, I find major issues with the foundation of this work, including the novelty and claims and do not recommend its publication in the journal of Nature Communications, in the current format.

Major comments:

1- The manuscript starts with a summary of DNA's capabilities and potentials as a data storage medium, which is fair and true. Then they describe the current DNA-synthesis methods as unscalable and unsustainable. This might be true in regard to phosphoramidite-based synthesis. There are in fact many environmental and cost issues with this method. However, it would only be fair if the authors mention the enzymatic (TdT-based) synthesis as well. The authors have cited the Lee et al. Nature Communications (2020) paper where enzymatic DNA synthesis is used for in-parallel data storage, exactly mentioning the promise of sustainability and scalability. And as we know, since the time this paper was published, there have been huge advances in the field of enzymatic DNA synthesis and multiple startups have even started commercializing enzymatically synthesized oligos/enzymatic synthesizers. So overall, I believe the authors need to discuss how their suggested approach would be superior to the state-of-the-art synthesis methods.

2- The authors claim that for the first time they use "greenly synthesized DNA registers". First off, I could not find how exactly they prepare the registers, except mentioning "... we used bacteria to synthesize the DNA registers used for DMOS", which is quite vague and rather unscientific. And second, the use of word "green" can be precarious, considering what was mentioned above about TdT-based synthesis. Nevertheless, even assuming that they've cloned a plasmid into E.coli cells and amplified a certain region off of it as the DNA registers, this is very much the same as what has been done in Tabatabaei et al. Nature Communications (2020) where the DNA registers are amplicons of the E.coli genome. It might also be worth mentioning the need for synthetic gRNAs, which must be synthesized, or transcribed from synthetic coding DNA, and in any case, are more difficult to obtain and maintain compared to synthetic DNA.

3- In the Conclusion section, the authors outline: "DMOS has the potential to bypass some challenges of conventional DNA based data storage including generation of toxic wastes, scalability, editability, and rewritability". I am wondering how editability and rewritability can be applied to this system. In other words, a dC deaminase in being used and dC's in the target sequence are converted to dT's. Do the authors suggest a method of converting the dT's back to dC's, as a way of erasing the recoded data and enabling rewriting? Or do they propose another approach to edit and rewrite information in DMOS?

Later they mention: "... in most systems DNA is chemically synthesized, user needs access to DNA synthesizer machines in order to write a new data, and editing of the recorded data requires the synthesis of the whole or significant portions of the DNA pools." Rewriting can be done via PCR, site directed mutagenesis, or ligation+ digestion of a specific portion of a pool and does not necessarily require the regeneration of the data via de novo synthesis. The authors could demonstrate an example of editing and rewriting recorded data with the DMOS system and compare it with the existing methods. In the same paragraph, a few examples of data storage in DNA nanostructures have been named. As a suggestion, a table comparing the densities and cost effectiveness of DMOS vs. these methods as well as synthesis-based platforms could help the reader understand the advantages and drawbacks better.

Minor Comments:

The overall writing can be improved in terms of the flow and grammar. Figures are in poor quality, specifically the ones including illustrations, which makes it hard to read the labels.

Reviewer #2:

Remarks to the Author:

In this manuscript, Sadremomtaz et al. presents a novel method of storing digital data using DNA as the medium and CRISPR-Cas9 base-editing technology to write the data. The authors highlight the growing demand for storage solutions as digital information production increases exponentially. They critique the current de-novo DNA synthesis methods for DNA-based memory technologies, stating these methods are not scalable or environmentally sustainable due to the costs and large amounts of toxic waste produced. The authors propose a new DNA storage system called "DNA Mutational Overwriting Storage" (DMOS) that uses combinatorial, addressable, orthogonal, and independent in vitro CRISPR base-editing reactions to overwrite data on pre-existing DNA sequences, or "DNA tapes". These tapes are synthesized using green, environmentally friendly methods (e.g. in-vivo plasmid replication).

Overall, the paper does well in integrating two novel fields of research – DNA data storage and CRISPR gene editing. The concept is compelling. My concerns and comments are relatively minor:

In the main text, the authors state the costs of storing the 5 minute youtube video as \$7M, but in the SI the figure is \$382M. Also regarding the cost, are the authors using costs from the highest throughput scale of synthesis? I.e. they cite a number from IDT, but at what scale is this? Is TWIST less expensive at scale?

It's not clear to me how the barcode-based addressing scheme is less scalable than the domain permutation strategy? It seems like the opposite would be true. E.g. with just a 25 bp barcode address 10^{15} barcodes are possible, outpacing the permutation-based barcodes?

Further to the addressing question, the sequence barcode-based approach would not require de novo synthesis of new registers, just PCR-based address appending. In contrast to the domain permutation based address which would require de novo synthesis and/or assembly of each different register.

Does the Bayesian classifier take into account different thresholds for different domains? I could see there might be sequence-dependent mutation efficiencies.

Where do the majority of the errors come from? The writing process? Could different Cas9 incubation times/conditions increase writing efficiency/off-targets? Or are the errors from nanopore sequencing? Would be good to discuss..

It would also be helpful for the authors to discuss the need for de novo synthesis of the gRNA. Could the authors provide an estimate of what their current cost per bit is using their system? Also, writing time/bit? These would be a helpful benchmarks to the field of DNA data storage.

Dear Dr. Cloney, editors and reviewers of Nature Communications,

We greatly appreciate the time and effort that you and the reviewers have invested in reviewing the manuscript and providing constructive feedback. We believe the feedback has helped us to improve the quality of the work.

We have provided a point-by-point response to the comments we have received. The reviewers' comments are in *[italic]*, and our responses are *[color-coded]*.

Reviewer 1

In this manuscript, Sadremomtaz et al. propose a DNA-based data storage architecture named 'DMOS', which uses dsDNA fragments with barcoded "domains" and a guided base editor (which is a dCas9+APOBEC3A) to record data. Basically, the enzyme fusion is guided towards a predetermined target sequence, and switches dC bases to dU and subsequently dTs after a resolving reaction (an exonuclease treatment followed by PCR). The recorded information is then read out by nanopore sequencing. The method is then used to record 1250 bit of data. While the design and experimental procedure are technically sound, I find major issues with the foundation of this work, including the novelty and claims and do not recommend its publication in the journal of Nature Communications, in the current format.

Major comments:

1- The manuscript starts with a summary of DNA's capabilities and potentials as a data storage medium, which is fair and true. Then they describe the current DNA-synthesis methods as unscalable and unsustainable. This might be true in regard to phosphoramidite-based synthesis. There are in fact many environmental and cost issues with this method. However, it would only be fair if the authors mention the enzymatic (TdT-based) synthesis as well. The authors have cited the Lee et al. Nature Communications (2020) paper where enzymatic DNA synthesis is used for in-parallel data storage, exactly mentioning the promise of sustainability and scalability. And as we know, since the time this paper was published, there have been huge advances in the field of enzymatic DNA synthesis and multiple startups have even started commercializing enzymatically synthesized oligos/enzymatic synthesizers. So overall, I believe the authors need to discuss how their suggested approach would be superior to the state-of-the-art synthesis methods.

We thank the reviewer for pointing out the new technologies for enzymatic DNA synthesis, which aligns with our concern for environmental issues. We added more content to the manuscript (Page 2, line 9), and highlighted the advantages of our approach, which utilizes a pre-existing blank pool of DNA tapes that can be massively generated through bacterial cloning (Page 2, line 27).

2- The authors claim that for the first time they use "greenly synthesized DNA registers". First off, I could not find how exactly they prepare the registers, except mentioning "... we used bacteria to synthesize the DNA registers used for DMOS", which is quite vague and rather unscientific. And second, the use of word "green" can be precarious, considering what was mentioned above about TdT-based synthesis. Nevertheless, even assuming that they've cloned a plasmid into E.coli cells and amplified a certain region off of it as the DNA registers, this is very much the same as what has been done in Tabatabaei et al. Nature Communications (2020) where the DNA registers are amplicons of the E.coli genome. It might also be

worth mentioning the need for synthetic gRNAs, which must be synthesized, or transcribed from synthetic coding DNA, and in any case, are more difficult to obtain and maintain compared to synthetic DNA.

We appreciate the precision of the reviewer regarding the generation of the DNA registers. We have updated the manuscript and clarified that we designed the DNA registers containing the 16 domain locations, that were chemically synthesized once, and amplified registers using bacterial cloning (Page 4, line 42) For the writing mechanism we require a library composed of 16 synthetic gRNAs (Page 11, line 28-35).

3- In the Conclusion section, the authors outline: "DMOS has the potential to bypass some challenges of conventional DNA based data storage including generation of toxic wastes, scalability, editability, and rewritability". I am wondering how editability and rewritability can be applied to this system. In other words, a dC deaminase is being used and dC's in the target sequence are converted to dT's. Do the authors suggest a method of converting the dT's back to dC's, as a way of erasing the recorded data and enabling rewriting? Or do they propose another approach to edit and rewrite information in DMOS?

We thank the reviewer for pointing out this issue. The current state of DMOS does not handle editability or re-writing. We revised the manuscript to reflect on the reviewer's comment (Page 9, line 29). Additionally, we have discussed the potential of employing reversal mutations by using Adenosine base editors (Page 10, line 10) and have added more content to discuss the matter (Page 10, lines 3-37).

4- Later they mention: "... in most systems DNA is chemically synthesized, user needs access to DNA synthesizer machines in order to write a new data, and editing of the recorded data requires the synthesis of the whole or significant portions of the DNA pools." Rewriting can be done via PCR, site directed mutagenesis, or ligation+ digestion of a specific portion of a pool and does not necessarily require the regeneration of the data via de novo synthesis. The authors could demonstrate an example of editing and rewriting recorded data with the DMOS system and compare it with the existing methods. In the same paragraph, a few examples of data storage in DNA nanostructures have been named. As a suggestion, a table comparing the densities and cost effectiveness of DMOS vs. these methods as well as synthesis-based platforms could help the reader understand the advantages and drawbacks better.

We appreciate the reviewer's observation. We corrected the claims regarding data edit and emphasized on the fact that using DMOS one can write data on blank templates without requiring the use of specialized DNA synthesis each time a new data is being saved (Page 9, line 10).

5- The overall writing can be improved in terms of the flow and grammar. Figures are in poor quality, specifically the ones including illustrations, which makes it hard to read the labels.

We reviewed the manuscript vigorously and made changes to improve the flow and quality of the manuscript. Additionally, we improved the quality of the Figures.

Reviewer 2

In this manuscript, Sadremomtaz et al. presents a novel method of storing digital data using DNA as the medium and CRISPR-Cas9 base-editing technology to write the data. The authors highlight the growing demand for storage solutions as digital information production increases exponentially. They critique the current de-novo DNA synthesis methods for DNA-based memory technologies, stating these methods are not scalable or environmentally sustainable due to the costs and large amounts of toxic waste produced. The authors propose a new DNA storage system called "DNA Mutational Overwriting Storage" (DMOS) that uses combinatorial, addressable, orthogonal, and independent in vitro CRISPR base-editing reactions to overwrite data on pre-existing DNA sequences, or "DNA tapes". These tapes are synthesized using green, environmentally friendly methods (e.g. in-vivo plasmid replication).

Overall, the paper does well in integrating two novel fields of research – DNA data storage and CRISPR gene editing. The concept is compelling. My concerns and comments are relatively minor:

1- In the main text, the authors state the costs of storing the 5 minute youtube video as \$7M, but in the SI the figure is \$382M. Also regarding the cost, are the authors using costs from the highest throughput scale of synthesis? I.e. they cite a number from IDT, but at what scale is this? Is TWIST less expensive at scale?

We appreciate the attention of the reviewer to this calculation. We verified the calculations are correct according to recent TWIST quotes (\$7M) and edited the manuscript and the supplementary materials accordingly (Page 2, line 3).

2- It's not clear to me how the barcode-based addressing scheme is less scalable than the domain permutation strategy? It seems like the opposite would be true. E.g. with just a 25 bp barcode address 10^{15} barcodes are possible, outpacing the permutation-based barcodes?

We thank the reviewer for pointing out this question. The reviewer is correct with respect to the greater capacity that is possible with barcode-based addressing. In our work we have used nanopore sequencing to read the data, and the data recovery was more computationally costly when we used barcode-based addressing. We have edited the manuscript accordingly to clarify our statements (Page 4, line 27).

3- Further to the addressing question, the sequence barcode-based approach would not require de novo synthesis of new registers, just PCR-based address appending. In contrast to the domain permutation based address which would require de novo synthesis and/or assembly of each different register.

*We thank the reviewer for this observation. We incorporated this clarification in our manuscript, indicating that the initial generation of the 48 registers required *de novo* synthesis, and then we amplified them using bacteria (Page 4, line 42).*

4- Does the Bayesian classifier take into account different thresholds for different domains? I could see there might be sequence-dependent mutation efficiencies. Where do the majority of the errors come from? The writing process? Could different Cas9 incubation times/conditions increase writing efficiency/off-targets? Or are the errors from nanopore sequencing? Would be good to discuss.

We thank the reviewer for these questions. Our first Bayesian classifier uses one specific threshold for each domain, the second Bayesian classifier takes the position of the bases into consideration. (Supplementary Page 12, line 1). The errors come from mutation efficiency, number of cytosines in the

target region and nanopore sequencing. However, we cannot determine the percentage of the error contribution of each case.

5- It would also be helpful for the authors to discuss the need for de novo synthesis of the gRNA

We thank the reviewer for this observation. We clarified this topic in our manuscript (Page 11, line 29-35).

6- Could the authors provide an estimate of what their current cost per bit is using their system? Also, writing time/bit? These would be a helpful benchmarks to the field of DNA data storage.

We thank the reviewer for pointing out this question. Estimating the true cost per bit for a DMOS reaction is difficult from our reported work as this was not our initial aim: we performed these enzymatic reactions entirely using commercially-available biological products and so the resulting cost/reaction was quite expensive, but we expect the precise cost will lower significantly as all the enzymatic components can/will be prepared at scale in-house and the size of individual reactions components are scaled down, as DMOS reactions are amenable to being performed using microfluidic systems. The price / bit also depends strongly on scale: as mentioned in the manuscript, once individual registers have been synthesized, they can be regenerated “infinitely”, so after an initial investment in DNA synthesis of a register, the price per register is minimal (cost of PCR or dsDNA purification) for all subsequent reactions and users. Nanopore sequencing using Oxford Nanopore technologies can also be orders of magnitude less expensive than the illumina sequencing that is necessary for synthesis-based methods, with multiplexed sequencing in principle bringing the price per run to \$10/sample (<https://store.nanoporetech.com/us/native-barcoding-kit-96-v14.html>; accessed August 7, 2023). Taken together, we expect the cost of DMOS data storage and retrieval can ultimately be made favorable compared to existing methods of nucleic acid memory.

With respect to write time / bit, again at this point we did not optimize for this consideration, but all reactions can be performed in parallel and automated (we used a pipetting robot here), so at the moment we expect that writing time / bit will scale linearly or sub-linearly with respect to the number of bits written, and will improve as automation and microfluidic fluid handling is achieved at scale.

Reviewers' Comments:

Reviewer #1:

Remarks to the Author:

The revised manuscript has improved technically and the writing/illustrations look much better. I'd like to thank the authors for addressing my questions.

Reviewer #2:

Remarks to the Author:

The authors have mostly addressed my concerns, though I would suggest adding some of the topics that were suggested to be discussed added to the manuscript (e.g. sources of error, and costs). Brevity is fine.